



# Global flood hazard map and exposed GDP comparison: a China case study

Jerom P.M. Aerts[1], Steffi Uhlemann-Elmer[2], Dirk Eilander[3,4], and Philip J. Ward[3]

[1] Water Resources Section, Faculty of Civil Engineering and Geosciences, Delft University of Technology, Delft, The Netherlands
[2] Aspen Insurance Ltd, Zurich, Switzerland
[3] Institute for Environmental Studies, Vrije Universiteit Amsterdam, Amsterdam, The Netherlands
[4] Deltares, Delft, The Netherlands

*Correspondence to*: Jerom P.M. Aerts (j.p.m.aerts@tudelft.nl)

**Abstract**

Floods are among the most frequent and damaging natural hazard events in the world. In 2016, economic losses from flooding amounted to $56 bn globally, of which $20 bn occurred in China (Munich Re, 2017). National or regional scale mapping of flood hazard is at present providing an inconsistent and incomplete picture of floods. Over the past decade global flood hazard models have been developed and continuously improved. There is now a significant demand for testing of the global hazard maps generated by these models in order to understand their applicability for international risk reduction strategies and for reinsurance portfolio risk assessments using catastrophe models. We expand on existing methods for comparing global hazard maps and analyse 8 global flood models (GFMs) that represent the current state of the global flood modelling community. We apply our comparison to China as a case study and, for the first time, we include industry models, pluvial flooding, and flood protection standards in the analysis. We find substantial variability between the flood hazard maps in modelled inundated area and exposed GDP across multiple return periods (ranging from 5 to 1500 years) and in expected annual exposed GDP. For example, for the 100 year return period undefended (assuming no flood protection) hazard maps the percentage of total affected GDP of China ranges between 4.4 % and 10.5 % for fluvial floods. For the majority of the GFMs we see only a small increase in inundated area or exposed GDP for high return period undefended hazard maps compared to low return periods, highlighting major limitations in the models' resolution and their output. The inclusion of industry models which currently model flooding at higher spatial resolution, and which additionally include pluvial flooding, strongly improves the comparison and provides important new benchmarks. Pluvial flooding can increase the expected annual exposed GDP by as much as 1.3% points. Our study strongly highlights the importance of flood defenses for a realistic risk assessment in countries like China that are characterized by high concentrations of exposure. Even an incomplete (1.74% of area of China) but locally detailed layer of structural defenses in high exposure areas reduces the expected annual exposed GDP to fluvial and pluvial flooding from 4.1 % to 2.8 %.



# 1 Introduction

Floods are one of the most frequent and most devastating kinds of natural disasters. Between 1980 and 2016, floods caused 23 % of overall economic losses and 14 % of fatalities due to natural hazards worldwide (Munich Re, 2017). In 2016, economic losses from flooding amounted to $56 bn globally. Understanding the risk of natural hazards, including flood risk, has therefore been identified as a priority in recent international risk reduction frameworks, such as the Sendai Framework for Disaster Risk Reduction (UNISDR, 2015).

In recent years, significant scientific efforts have been carried out to develop global flood risk models (GFMs) (Teng et al., 2017). In terms of river flooding, these have examined current flood risk at the global scale (e.g., Winsemius et al. 2013) as well as future flood risk due to changes in: hazard, as a result of climate change (Alfieri et al. 2015; Alfieri et al. 2016; Arnell and Gosling 2016; Hirabayashi et al. 2013; Kundzewicz et al. 2014; Ward et al., 2017; Winsemius et al. 2015); exposure, due to increasing population, wealth, and urbanisation (Hallegatte et al., 2013; de Moel et al., 2015); and vulnerability (Jongman et al., 2015). To date, attention has especially been paid to developing global flood hazard maps. These maps indicate the severity of the hazard for different exceedance probabilities across the globe. The hazard severity is generally expressed in terms of flood extent and flood depth, on a raster grid with resolutions ranging from 1 arc second to 32 arc seconds. The GFMs that are used to create these flood hazard maps are simplified global scale models of surface water flows that are driven by regional or global climate models or rely on gauged discharge or (gauged) precipitation datasets (Sampson et al., 2015). The development of these models has been facilitated by advances in satellite data, numerical algorithms, computing power, and coupled modelling frameworks (Ward et al., 2015). The key advantage of GFMs compared to regional or national flood models is their global scale, which means that flood hazard maps are now available in data-poor areas that previously lacked hazard maps (Hagen and Lu, 2011).

Despite these recent advances, several major challenges still exist. For example, Ward et al. (2015) discuss the quality of elevation data, accuracy of boundary conditions used to force inundation models, and the knowledge of river morphology, among other things. Bernhofen et al. (2018) also discuss the importance of forcing boundary conditions, especially input flow, as well as the influence of morphological features, such as floodplain size and the steepness of the terrain. Another major challenge for GFMs is to account for the impact that structural flood defenses have on flood hazard, especially in regions with high protection standards.

Due to the aforementioned challenges, and the growing number of GFMs, there is now a significant demand for comparing the outputs of different models and assessing their accuracy. This helps in understanding the applicability of GFMs for developing international risk reduction strategies and for their use in reinsurance and insurance portfolio risk assessments. Several such studies have been carried out by comparing or investigating a certain model component (e.g. global hydrological model, river routing model, and model resolution) in the GFM framework. For example, Schellekens et al. (2017) conducted an inter-model agreement assessment from 10 GHMs based on the signal-to-noise ratio in monthly mean anomalies of evapotranspiration, runoff, root zone soil moisture and precipitation. The agreement of the GHMs was found to be low in





snow-dominated regions and tropical rainforest or monsoon areas and high in temperate areas. A study by Zhao et al. (2017) assessed the ability of GHMs with native routing schemes to capture the timing and amplitude of river discharge. The results were compared to the use of a dedicated global river routing model, CaMa-Flood. Generally the use of CaMa-Flood improved the accuracy of simulating peak river discharge. Mateo et al. (2017) investigated the applicability of a GFM at higher spatial resolutions by validating it against a large past flood event in Thailand. They found that validation results improved with higher

spatial resolution if multiple downstream connectivity is represented in the river routing model.

Rather than testing and investigating a certain model component of GFMs, Trigg et al. (2016) compared flood hazard maps from 6 different GFMs for the African continent. The study compared the inundated area across hazard maps for multiple return periods and assessed how this translates into differences in exposed gross domestic product (GDP) and exposed population. They found large differences; for example over the continent of Africa there is around 60 % to 70 % of

disagreement between the GFMs in terms of inundated area. These differences are mainly present in deltas, arid climate zones, and wetlands. This study did not examine the effect of flood protection standards on flood hazard, or floods with a pluvial origin. The study concludes that in order to increase the quality of GFMs there is a demand for more inter-comparison studies and stresses the importance of the inclusion of industry models. In reply, Bernhofen et al. (2018) validated the same 6 GFMs in Africa. The best individual models performed at an acceptable level compared to observations. Further findings were that

models forced by river gauged flow data outperform models forced by climate reanalysis data. Contrary to previous studies, no relationship was found between performance and model spatial resolution. In a follow-up study, Hoch and Trigg (2019) proposed a global flood model validation framework. The aim of this framework is to understand the drivers of deviations between GFMs by providing standard forcing data, validating and benchmarking model results, and by sorting and indexing reference output. This framework is in line with the currently developed eWaterCycle II platform, which provides the above-

mentioned principles for the global hydrological modelling community (www.ewatercycle.org, Hut et al. (2018)).

In this study, we expand upon the existing work of global flood model inter-comparison studies. The main aim is to carry out a comprehensive comparison of flood hazard maps from 8 GFMs for the country of China, and assess how differences in the simulated flood extent between the models lead to differences in simulated exposed GDP and expected annual exposed GDP. This is carried out by addressing the variation in different model structures and the variability between flood hazard map. Our

comparison uses both publicly available GFMs (GLOFRIS, ECMWF, CAMA-UT, JRC, and CIMA-UNEP) as well as industry models (Fathom, KatRisk, and JBA) that are applied within the wider re-insurance industry. It is the first comparison study to include industry models, the pluvial flood component, and the role of flood protection on the flood hazard and exposure.

China is selected as our case study area because it poses many challenges to flood modelling: data scarcity; a variety of flood mechanisms spanning many climatic zones; complex topography; strong anthropogenic influence on the flood regimes, for

example through river training; and a very high concentration of exposure. Moreover, China is prone to severe flood events. For example, in June 2016 alone more than 60 million people were affected by floods, resulting in an estimated damage of $22 bn (CRED 2016).



This paper is set up as follows. In Section 2, we describe the data and models used in this study. In Section 3, we describe the (statistical) methods applied to compare the data from the various models. In Section 4, we present and discuss the results, 100 examining differences in flood hazard, exposed GDP, and expected annual exposed GDP between GFMs, the influence of incorporating flood protection, and model agreement. Conclusions and outlook are provided in Section 5. In the supplement S1, we provide a detailed overview of the models and data used.

## 2 Description of flood hazard maps and models

We compare flood hazard maps for different return periods from 8 different GFMs, namely: CaMa-UT (Yamazaki et al. 2011, 105 2014, 2015), GLOFRIS (Ward et al. 2013; Winsemius et al. 2013), JRC (Dottori and Todini, 2011), ECMWF (Balsamo et al., 2015), Fathom (Sampson et al., 2015), CIMA-UNEP (Rudari et al. 2015), KatRisk (contact KatRisk for a technical report) and JBA (contact JBA for a technical report). The native flood hazard map outputs of each GFM were acquired between November 2017 and May 2018. Data were downloaded or requested in their original published format (at the time of the study) and no bespoke or post-processed maps were requested. The flood hazard maps are undefended (CIMA-UNEP has readily built in 110 flood protection, Fathom and JBA provided defended hazard maps, section 2.5), fluvial floods only and fluvial with pluvial floods combined (Fathom, KatRisk, and JBA), the so called combined flood hazard maps. The hazard maps cover return periods (RPs) ranging from 5 to 1500 years and the native flood hazard map output resolutions range from 1 arc second to 32 arc seconds.

## 2.1 Model structures

From the 8 GFMs, we identified two groups based on the model structure described in Trigg et al. (2016): the cascade model structure (CaMa-UT, GLOFRIS, JRC, ECMWF, and KatRisk) and the gauged flow model structure (Fathom, CIMA-UNEP, and JBA). An overview of the modelling chain of both model structures is shown in Figure 1 and further explained in sections 2.2.1 and 2.2.2. A concise description of the cascade model structure is provided by Winsemius et al. (2013) and by Sampson et al. (2015) for the gauged flow model structure.

The general model input data used by the GFMs (i.e. river network datasets, digital representations of the earth's surface like digital elevation models (DEM), digital terrain models (DTM) or digital surface models (DSM)) vary in type, resolution and corrections applied. CaMa-UT, GLOFRIS, JRC, ECMWF, CIMA-UNEP, Fathom and KatRisk use the HydroSHEDS river network (Lehner and Grill, 2013) and SRTM3 DEM (Farr et al., 2007) at either 3 or 30 arc seconds. Urban and vegetation bias corrections are applied before use. Additionally, KatRisk applies an algorithmic filtering to clean the DEM and uses manual 125 correction to remove blockages of flow pathways. The JBA method uses the Intermap Technologies Inc. NEXTMap WORLD30 digital surface model (DSM) for China. The DSM provides global coverage at 1 arc second resolution. On a global scale, the JBA method uses a bare earth DTM to complement the DSM. The JBA method derives the river network from elevation data and applies extensive validation and correction before use.



The summary of model characteristics in Table 2 shows the model structures, climate forcing datasets, and GHM (when applicable), the name and type of river routing models, considered catchment size, type of digital elevation model, downscaled model resolution, and the native output resolution of the flood hazard maps.

### 2.1.1 Cascade model structure

The defining characteristics of the cascade model structure are the use of climate forcing input datasets for the GHMs. River routing models then calculate the continuous river flow along river networks, calculating river and floodplain inundation dynamics. This is followed by flood frequency analysis (FFA), which determines flood depth and extent for a given RP or the flood volume in the case that downscaling is required.

Following the numeration of Figure 1, the cascade modelling chain starts with:

[1] Climate forcing datasets that provide precipitation, temperature, and in some cases potential evapotranspiration time series as input for GHMs. The datasets (JRA-25, EU-WATCH, ERA-INTERIM, EC-EARTH) vary in modelled time period, time step, resolution, and atmospheric processes. The modelled time periods range from 1979 up to present day, with all periods spanning more than 30 years to avoid bias by inter-decadal variability. The time step of the climate forcing datasets is 6 hourly and the horizontal resolutions range between 80 km to 1.125 degrees. The KatRisk model uses gridded daily precipitation observations from the US National Weather Service's Climate Prediction Center (CPC) to establish rainfall-runoff relationships in combination with the ERA-Interim dataset that provides other atmospheric variables used to estimate evapotranspiration (like wind speed, radiation, and temperature).

[2] The GHMs calculate the surface and atmosphere interactions. GHMs vary in modelled processes, time steps and resolution. The modelled processes mainly deviate in how runoff, evapotranspiration, and snow schemes are executed. The time steps of the GHMs are hourly (CaMa-UT, JRC, ECMWF), 3 hourly (CIMA-UNEP), 6 hourly (KatRisk), or daily (GLOFRIS). The GHM resolutions range between 3 arc seconds (CIMA-UNEP, KatRisk), 0.1 degrees (CaMa-UT, JRC, ECMWF) and 0.5 degrees (GLOFRIS). The GHMs produce specific discharge along river networks, which is then passed through river routing models.

[3] A wide range of methods is used to model inundation dynamics. The complexities range from 2D flood volume redistribution (GLOFRIS), complex 2D sub-grid topography models (CaMa-UT, ECMWF), towards 2D hydrodynamic models (JRC, KatRisk). Main differences between the river routing models are the resolution and the formulation of the shallow water equations. The resolutions range from 3 arc seconds (KatRisk), 0.1 degrees (JRC), 0.25 degrees (CaMa-UT, ECMWF), to 0.5 degrees (GLOFRIS). The shallow water equations used for calculating the river routing are either local intertia (CaMa-UT, ECMWF), kinematic wave (GLOFRIS, JRC), or a unit hydrograph approach (KatRisk) where upstream and lateral inflow are treated as instantaneous inputs to a linear time-invariant model using the advection-diffusion equation as response function.

[4] The output of the global river routing model is used to estimate a time series of flood volume (GLOFRIS) or flood depth (CaMa-UT, JRC, ECMWF, KatRisk). Applying flood frequency analysis (FFA), annual maxima of local runoff and/or river





discharge are extrapolated to RPs beyond the observational space using extreme value distributions. All models use Gumbel extreme value, to estimate peak values for each RP.

[5] The resulting flood volumes or depths per computation cell are downscaled to increase the output resolution. Either the water level is downscaled (CaMa-UT, JRC, ECMWF) or the flood volume is redistributed to the resolution of the digital

elevation model (GLOFRIS). The KatRisk model does not require further downscaling. The resolutions are 3 arc seconds (CaMa-UT, ECMWF) and 30 arc seconds (JRC, GLOFRIS). The native output resolutions are 3 arc seconds (KatRisk), 18 arc seconds (CaMa-UT, ECMWF) and 30 arc seconds (JRC, GLOFRIS).

### 2.1.2 Gauged flow model structure

Following the numeration of Figure 1, models belonging to the gauged flow model structure use gauged discharge or gauged

precipitation datasets as input. The modelling approaches differ between using regionalization techniques that depend on upstream catchment characteristics (Fathom), models that need to be complemented by hydrologic simulations (CIMA-UNEP), to those that use empirical rainfall-runoff methods (JBA). Based on the output of these methods, the flood flow magnitude is calculated through flood frequency analysis for given RPs that force river routing models. The river routing models produce flood extents and flood depths for given RPs. The gauged flow models in this study do not require

downscaling.

[1] For the water volume input, the CIMA-UNEP and Fathom models use the Global Runoff Data Centre (GRDC, Germany) river discharge dataset as their main input of discharge observations. This dataset consists of more than 9500 stations that collect their data at daily and monthly intervals. Of these 9500 stations, only 39 are located in China. The Fathom model is complemented with the United States Geological Survey (USGS) stream gauge dataset. The JBA method uses the Climate

Research Unit (CRU) TS 3.2 (>4000 weather stations) (Harris et al., 2014) and Climate Forecast System Reanalysis (CFSR) v2 precipitation dataset (Saha et al., 2010), which respectively cover the period 1901 to 2011 and 1979 to 2009 with a monthly and daily temporal resolution. The CFSR data are calibrated using 25 rain gauges in China. For China, 170 river gauges are used to enable the modelling of empirical rainfall-runoff relationships to calculate river discharge.

[2] The CIMA-UNEP and Fathom models follow the assumption that inferences from data-rich catchments can be transferred

to data poor catchments. Discharge data are first pooled into homogeneous regions based on catchment descriptors of climate, upstream annual rainfall and catchment area, after which they are divided into the five classes of the Köppen-Geiger climate classification (Kottek et al., 2006; Sampson et al., 2015). Regional flood frequency curves are derived using the generalized extreme value distribution and are combined with the index flood to generate return period design flood hydrographs along the river network (Sampson et al. 2015; Smith et al. 2015).

The CIMA-UNEP model is complemented with hydrologic simulations using the EC-Earth climate forcing dataset and the continuum model to ensure that results are correct in data-scarce catchments. The JBA model does not require regression techniques as their precipitation datasets have global coverage.





[3 and 4] The flood hydrographs are then used to force river routing models that propagate the flow across digital elevation models, calculating flood depth and extent without the need for downscaling. As with the cascade models, the river routing
models of the gauged flow models vary in methods and complexity. JBA uses the RFlow model for all of the large river networks in China, except for the downstream end of the Pearl River (Guangzhou area) and the downstream end of the Yangtze River (Shanghai area), which are modelled with JFlow in a fluvial configuration. Small rivers (catchments < 500km$^2$) as well as surface water flooding are modelled using JFlow in a direct-rainfall configuration. The resolutions of the river routing models vary between 1 arc second (RFlow, JFlow), 3 arc seconds (CIMA-UNEP), and 30 arc seconds (Fathom). The shallow
water equations used for calculating the river routing are inertia (Fathom), Mannings equations (CIMA-UNEP), the combination of the Normal Depth and Mannings equations (JBA-RFlow model) and the full shallow water equations (JBA-JFlow model).

### 2.1.3 Pluvial flood modelling

In addition to fluvial floods, the JBA, Fathom, and KatRisk models also simulate pluvial floods. Fathom uses a "rain-on-grid"
method for rivers and catchments smaller than 50 km$^2$, where flow is generated by raining directly on the DEM at 3 arc seconds in order to calculate runoff. This method uses Intensity-Duration-Frequency (IDF) relationships to estimate the duration, intensity and frequency of extreme rainfall before applying the same regression techniques for extrapolation as with the fluvial component. The JBA method follows a similar approach by calculating IDF relationships at the centroid of each CFSR tile (0.5degree x 0.5degree). Kriging is used to interpolate between the tile centroids to create a continuous rainfall surface for
each RP and storm duration (three storm durations are included; 1, 3 and 24 hour). The JFlow routing model is run in this direct-rainfall approach to model the small rivers (<500 km$^2$) and surface waters. The KatRisk model uses daily precipitation from the Climate Prediction Centre dataset *(*Boulder, USA) to simulate rainfall over catchments smaller than 500 km$^2$. The precipitation dataset combines all available historical data sources for daily and sub-daily global coverage from 1979 to real-time, longer for monthly data. The data are checked for errors and to ensure spatial and temporal consistency. A 2D storage
cell (diffusive wave) model is used to calculate pluvial flood patterns. The runoff is distributed uniformly across a catchment and routed according to topography at 3 arc seconds. The flow (surface runoff fraction) is calibrated using river gauged discharge data.

### 2.2  Defended hazard maps and external flood protection layers

Of all global flood models considered in this study, three include options for considering the impact of structural flood defenses
on the hazard maps.
The CIMA-UNEP hazard maps are the only maps that contain a level of built-in flood protection, which cannot be removed. They incorporate flood protection standards by creating a defense ellipsoid around large cities, with the size being dependent on the GDP. All flooding within this ellipsoid is removed in post-processing and the defenses are assumed to fail above a





standard of protection of RP200. Hence, this also means that for the CIMA-UNEP model the undefended baseline hazard maps
are not available for this study.

Alongside the undefended hazard maps, Fathom also provided flood hazard maps with integrated flood protection. JBA further
provided a dataset of defenses (largely for dense urban areas) that can be superimposed on the flood hazard maps to create a
defended set of flood maps per return period.

To allow for comparison between the individual GFMs, we decided to include defenses only in a post-processing step using
not-built in layers of defenses, meaning that Fathom's defended maps were not used in this study. Section 3.5 describes the
post-processing step in more detail.

The two flood protection layers used in this study are: 1) a county level defenses layer, and 2) a city level defenses layer. The
first layer was created by Du (2018), and describes standards of protection (SoP) on an administrative county level covering
the whole of China. It can be considered as a kind of policy layer as it makes assumptions about the degree of protection based
on goods to be protected. This layer was developed by dividing counties into urban or rural areas. The urban area SoPs are
based on GDP and population datasets from the Chinese government. The GDP dataset is converted into a weighted population
dataset and is then combined with the population dataset to calculate the maximum urban protection for a given county. The
rural area SoP is based on the assumption that farmland is a key indicator for flood protection due to its importance for
providing food security for the large population of China. The area of farmland is derived from a governmental land use map
and is combined with the population dataset to calculate the maximum SoP for each county. The urban and rural areas within
the counties are then combined to create a nationwide layer of flood protection standards. The SoPs of the layer range from 10
in rural counties (western China) to 200 in urban counties (eastern China).

The second layer is the high resolution JBA Defended Areas flood protection layer and is from hereon in referred to as the city
level defense layer. The layer is a national layer that contains SoP polygons with a focus on urban areas. The defended areas
are determined using a variety of best available third party sources. Some of the defended areas were excluded by JBA as it is
likely that flooding might occur from surrounding undefended areas. The SoP attributed to each defended area is determined
from the local available data source. Where it was not known, the defended area was attributed with the SoP of either the
neighboring defense data or the regional average. In total, the layer covers only 1.74% of the area of China.

## 3 Methodology

We assess the agreement between the flood hazard maps of the 8 GFMs by calculating the inundated area for the whole of
China and by applying a model agreement index that calculates the agreement on inundation per grid cell. We include a GDP
layer to study how inundated area relates to exposed GDP, the amount of expected annual exposed GDP, and how model
agreement relates to agreement on the amount of exposed GDP. By including flood protection standards we can assess the
effects of these layers on the previous mentioned types of analyses, benefitting the knowledge of the importance of including



such layers in further studies. In addition, we ensure a fair and accurate comparison of the flood hazard through the use of a
data homogenization scheme.

### 3.1 Data homogenization

We acquired the undefended flood hazard maps of the global flood models (GFM) in their native output format. The difference
in resolutions and output formats require an initial homogenization of the data. Firstly, the hazard maps were masked to the
case study area extent. The extent includes continental China, excluding Hong Kong, Macau and Taiwan. Thirdly, we
disaggregated the hazard maps to a 3 arc-second resolution. The chosen resolution is a balance between minimizing the loss
of data quality while maintaining manageable file sizes and processing time. The disaggregation was conducted with the
nearest-neighbour resampling technique, meaning that a single 30 arc second grid cell is resampled to ten 3 arc second grid
cells with the same value. The Fathom and KatRisk model outputs did not require resampling as their hazard maps are native
at 3 arc seconds. The JBA flood hazard maps were aggregated to 3 arc seconds from their native 1 arc second hazard map
resolution. Fourthly, the hazard maps were converted from representing flood depth, when available, to flood extent by
changing all grid cell values larger than 0 to 1. This decision was made due to the lack of flood depth availability in all flood
hazard maps. Lastly, 'permanent' waterbodies were removed from the flood hazard maps. The GFMs disagree on the inundation
of lakes and rivers. To avoid a large positive bias in the hit rate we removed these 'neutral water bodies' from the hazard maps
using an independent dataset. The global surface water 1984-2015 dataset from the Joint Research Centre (Pekel et al., 2016)
was modified to represent 'neutral water bodies' as areas that are inundated 80 percent of the time or more during the 1984 to
2015 period. This percentage of occurrence ensures that permanent lakes and rivers are removed, whilst minimizing the
removal of floodplain inundation.

### 3.2 Inundation percentages

We compared the amount of inundated area between the different flood hazard maps with and without flood protection
standards. To accurately calculate the inundated area in $km^2$ we implemented the Haversine method (Brummelen, 2013). Using
this method we created a grid containing the accurate size in $km^2$ of each grid cell. Next, we divided the inundated area of the
flood hazard maps by the total land area of China to express the results as an percentage of inundated area of the total land
area of China.

### 3.3 Exposed GDP and Expected Annual Exposed GDP

The exposed GDP was calculated by overlaying the flood hazard maps with a gridded GDP layer created by Kummu et al.
(2018). This layer has a native resolution of 30 arc seconds and represents the year 2015. We first adjusted the resolution of
the GDP layer to 3 arc seconds using the bilinear resampling technique. Next, we multiplied the homogenized flood extent
hazard maps with the GDP layer to obtain the exposed GDP value for each inundated grid cell. The results were then divided
by the total GDP of China to express the exposed GDP as a percentage of the total GDP of China. In addition, we calculated





the expected annual exposed GDP (EAE-GDP) following the method of Apel et al. (2016). The EAE-GDP is the result of the flood event probability of exceedance ($P$) and its exposure ($E$).

$$EAE = \sum_{i=1}^{n} \Delta P_i \cdot \bar{E}_i \qquad (1)$$

$$\Delta P_i = P_{i+1} - P_i$$

$\quad \Delta \bar{E}_i = \frac{1}{2}(E_i + E_{i+1}),$

$\Delta P$ is the change in annual probability of exceedance where $P = \frac{1}{RP}$ (Triet et al., 2018). $E$ is the exposed GDP, $i$ the numerator of RP under consideration (with $i = 1$ representing RP5 in this study), and $n$ is the number of considered RPs. The RPs that were not represented by the individual GFMs were filled to ensure that the lack of especially low RP data does not distort the actual EAE. The data gaps were filled using linear interpolation and extrapolation for RP5 to RP1500 based on the exposed

GDP percentage results. This can have a large effect on the results of GFMs that lack lower RP flood hazard maps as they will likely have an overestimation of exposed GDP due to linear extrapolation.

### 3.4 Model Agreement Index

The model agreement index (MAI) was introduced by Trigg et al. (2016) as a measure for expressing model agreement on a grid cell level. We calculated the MAI for the RPs 20-25, 50, 100 and 500, because these are available for all 8 GFMs. A

distinction is made between the fluvial and combined hazard maps. Before MAI calculation, the binary hazard maps (data homogenization processes) were aggregated (stacked), resulting in grid cell values ranging from 0 to 7 for the fluvial hazard maps and grid cell values ranging from 0 to 3 for the combined hazard maps. KatRisk's maps produce fluvial and pluvial flood hazard combined and are therefore not included in the fluvial MAI calculation.

$$MAI = \frac{\sum_{i=2}^{n} \frac{i}{N} a_i}{A}, \qquad (2)$$

where $N$ is the number of models under consideration, $i$ the number of models in agreement, $a_i$ the inundated area for the number ($i$) of models in agreement, and $A$ is the total inundated area of all models under consideration.

The MAI formula in Eq. 2 has an output value between 0 (no agreement) and 1 (perfect agreement). The formula only takes into account inundated grid cells in order to avoid misrepresentation of model agreement. The large number of non-inundated grid cells would create bias due to a high hit rate. An example of a model agreement grid with MAI calculation is provided in

Table 3.

### 3.5 Defended hazard maps

We assess the influence of flood protection on the inundated area, exposed GDP, EAE-GDP, and MAI using two different types of defenses to reflect two typically used strategies for modelling structural defenses. A) a county level and largely policy based defense layer and B) a national layer defense layer with a focus on urban areas on a city-scale that delineates defenses

only in areas of highest exposure (described in section 2.2). The undefended hazard maps of all models considered in this study



were used. For the special case of the CIMA-UNEP flood hazard maps, which include a built-in defense layer, we still super-impose the defense layers. The defended flood hazard maps are created by masking areas that are protected for a given standard of protection (SoP). For example, a grid-cell that is inundated at RP100 and has a protection level of SoP100 is considered to be not inundated and is therefore masked in the flood hazard map.

## 4 Results and Discussion

### 4.1 Spatial distribution of floods

Figure 2 shows the RP100 flood extent for both fluvial (2a) and fluvial with pluvial flooding combined (2b) across China. Noticeable are the large inundated areas in the Xinjiang province of northwestern China, the northeastern provinces of Heilongjiang, Jilin, and Liaoning, as well as the large deltas located in the east. The latter consists of the large cities of Beijing and Shanghai (among others), and is therefore a region of high exposure.

### 4.2 Inundated area and flood protection

The comparison of inundated area (expressed as a percentage of the total land area of China) between different models is shown in Figure 3 (a-c). The figures show both the fluvial hazard maps and the combined hazard maps (fluvial and pluvial floods), with RPs ranging from 5 to 1500. Results are shown for the undefended layers (3a) and the defended layers (3b and c).

Focusing first on the undefended fluvial hazard maps in Figure 3a (solid lines), the predicted spread in percentage of inundated area ranges between 4.3 % and 9.8 % for RP20, and 5.8 % to 14.2 % for RP500. The CaMa-UT, GLOFRIS and JRC models show very similar results across RPs and generally low amounts of percentage of inundated area compared to the other GFMs. The ECMWF, Fathom and CIMA-UNEP models show similar results across RPs and moderate amounts of percentage of inundated area. JBA's maps produce the highest percentage of inundated area across all RPs.

The differences and similarities in results cannot be explained by differences in model structure alone. The GFMs with the closest resemblance in model structure and model components (Table 2) are the CaMa-UT and ECMWF models and the results of both models differ up to a factor 2. These models use different climate forcing datasets (JRA Reanalysis, ERA-Interim) and GHMs (MATSIRO-GW, HTESSEL), the rest of the model structure is similar. From the resemblance in model structures of the CaMa-UT and ECMWF models it can be inferred that the difference in global climate forcing and GHM have large effects on the percentage of inundated area.

The difference in inundated area between low and high RPs is small for the majority of models (figure 3a), with the exception of the Fathom and JBA models. The CaMa-UT and ECMWF models show a similar increment across the different RPs (though there is a large absolute difference between the two models), which is possibly caused by the similar output resolution (18 arc seconds) and considered catchment size (500 km$^2$). GFMs with higher output resolutions and smaller considered catchment sizes tend to have larger increments between different RPs in the results, such as the JBA model. Moreover, the high output



resolution and the inclusion of catchments of very small size in the JBA model are likely the reason for the hazard maps to predict inundation percentages significantly higher than the other models.

For the 6 GFMs (excluding JBA and KatRisk) that were used in the study of Trigg et al (2016), the undefended fluvial hazard map percentages of area inundated in our study for China are similar to those found in Africa by Trigg et al. I.e., the inundation percentages range between 3 % to 8.2 % for RP20 and 3.5 % to 9.5 % for RP500. More general similarities between the two studies are found. For example, the percentage of inundated area is highest for the ECMWF and Fathom models in both studies. However, the results based on the CIMA-UNEP model are very different, with a relatively high percentage of inundation (double) in our study compared to the study of Trigg et al. (2016). However, it should be noted that the output resolution of the CIMA-UNEP hazard maps used in our study (32 arc seconds or ~1km) is lower than the resolution used by Trigg et al. (2016) (3 arc seconds or ~90m). Rudari et al. (2015) tested the role of output resolution on the hazard maps of CIMA-UNEP. They found that aggregating data from 3 arc seconds to 32 arc seconds has major implications; for 22 case study areas investigated in East-Asia, they found an increase of inundation amount by a factor 2 on average. Their findings correspond well with the difference in CIMA-UNEP results between both studies and further underline the large influence of output resolution on flood hazard maps.

The combined fluvial and pluvial hazard maps shown in Figure 3a (Fathom, KatRisk, and JBA models, dashed lines) show less variation for a given RP than the undefended fluvial hazard maps. The values vary between 8.0 % and 10.5 % for RP20, and 15.2 % to 17.7 % for RP500. The difference in inundated area between the JBA fluvial and combined hazard maps is relatively stable across increasing RPs. However, this is not the case for the difference between the Fathom fluvial and combined hazard maps, which increases as the RP increases. The higher amounts of inundation percentage due to the addition of pluvial floods (2 % points Fathom and 0.9 % points JBA for RP100) highlight the importance of including pluvial floods in flood hazard assessments at a large scale.

Next, we examine the defended flood hazard map results shown in Figures 3b-c. The defended county level flood hazard map results in Figure 3b are based on the assumption of complete protection against RP10 (rural areas) and up to RP200 (in urban areas), and no protection against RP250 floods and higher. The results show percentage of inundated area for RP20 ranging between 0.2 % and 1.5 %. The effect of including flood protection is largest for low RPs and becomes smaller with increasing RP. The results for RP100 vary between 4.4 % and 12.7 %. Compared to the undefended hazard maps the spread of results is reduced from 6.2 % points to 1.3 % points for RP20 and from 8.8 % points to 8.3 % points for RP100. The small difference between undefended and defended county level maps at RP100 is explained by the presence of flood protection in the economically prosperous and densely populated counties in eastern China, leaving more counties prone to flooding.

The defended city level hazard map results in Figure 3c do not assume complete protection against a given RP flood. The results are similar to the results of the undefended flood hazard maps because of the coverage of 1.74 % of China for this flood protection layer.



### 4.3 Exposed GDP and flood protection

The exposed GDP results (expressed as percentage of the total GDP of China) for the fluvial and combined hazard maps are shown in Figures 3d-f, for RPs ranging from 5 to 1500 years, with and without flood protection. Results for the undefended exposed GDP (Figure 3d) vary between 13.9 % and 27.8 % for RP20 and between 17.9 % and 33.4 % for RP100. Multiple similarities are found between the inundated area (Figure 3a) results and the exposed GDP (Figure 3d) results. The CaMa-UT, GLOFRIS and JRC models have the lowest percentages for both types of results per RP. Similarly, the combined hazard maps

of the KatRisk, Fathom and JBA models have the highest percentages for both types of results per RP. The main difference is for the ECMWF model, which has the highest percentages of exposed GDP between RP5 and RP100 as this is different from the inundated area results in which the inundated area is close to the average of all GFM results. Additionally, the Fathom model produces relatively low exposed GDP percentages compared to the fluvial percentage of inundated area, which were close to the average. These results depict that a high amount of inundated area does not necessary lead to a high amount of

exposed GDP and vice versa.

The high exposed GDP percentages of the ECMWF model are caused by the inundation of densely populated deltas in eastern China. This illustrates that inundated area alone does not give an adequate representation of the difference between models in terms of their use for assessing the impacts of floods. This is further illustrated by the relatively low exposed GDP percentages of the Fathom model, as compared to the high percentages of inundated area for that model, which is due to simulated

inundation in large parts of the sparsely populated regions of western China. The CIMA-UNEP results show a large increase in exposed GDP percentage between RP500 and RP1000 of 12.1 % points, which is due to flood protection assumed to being exceeded at this point and resulting in a major jump from the incorporated flood protection surrounding large cities at SoP200 and lower. This effect is larger for the exposed GDP than the inundated area assessment.

The defended county level exposed GDP results in Figure 3e vary between 0.1% and 0.2% for RP20 and between 8.8% and

17.6% for RP100. Compared to the undefended exposed GDP results (Figure 3d), the effect of including county level flood protection standards is larger for exposed GDP than inundated area. Generally, the variability between models in exposed GDP is very small between RP20 and increases towards RP100. At RP250 and higher the variability of results increases more due to floods exceeding the design values of the defenses for the large cities (where GDP is concentrated) in the delta areas. This has a larger effect on the exposed GDP of the fluvial hazard maps of the CaMa-UT, GLOFRIS, JRC, and Fathom models than

on the combined hazard maps of KatRisk, Fathom and JBA models.

The results of the city level defended exposed GDP in Figure 3f vary between 9.4 % and 18.5 % for RP20 and between 17.3 % and 32.5 % for RP100. Contrary to the small effect of county level defenses on the inundated area results, the impact is large for the exposed GDP results in respect to the small coverage of China (1.74 %). For example, the ECMWF model has a lower exposed GDP of 15.8 % for the city defended scenario as compared to 27.8 % for the undefended scenario at RP5. The

city defended results show less variability for the lower RPs than for the undefended exposed GDP. The variability among the





GFMs increases between RP50 and RP100 from 9.6 % to 15.2 % because the highest assumed level of flood protection for this layer is RP100.

The city level defenses reduce the spread of exposed GDP estimates drastically by a flood protection layer that covers only 1.74 % of China. This highlights the importance of including locally detailed flood protection data for the correct representation
of exposed GDP. Adding information from a policy layer can further improve the risk assessment on a country wide scale, but needs careful validation of the uniform per county total protection assumptions. Also, ideally, flood protection standards are already incorporated within the river routing models of the various GFMs instead of incorporation during post-processing.

### 4.4 Expected Annual Exposure

The expected annual exposed GDP (EAE-GDP) results shown in Table 4 are expressed as a percentage of the total GDP of
China. Generally, these results reflect the findings of the per RP comparison in the previous sections. The CIMA-UNEP model simulates much lower EAE-GDP than the other models for the undefended and defended county level EAE-GDP, which is due to the large difference in inundation percentages, caused by incorporated flood protection, between RP25 and RP50. Extrapolation of these results to RP5 leads to very low exposed GDP percentage estimates and therefore results in a low EAE-GDP value. This is not the case for the defended county level EAE-GDP due to all models agreeing on low amounts of exposed
GDP for RP20 and RP25. The agreement between GFMs causes the defended county level variation to be small, at 0.29 % point. To place these results in perspective, we compare against the annual average losses (AAL) based on EM-DAT data (CRED EM-DAT (Feb. 2015)) for the period 1990 to 2014. The AAL is estimated to be 0.20 % of the national GDP. The defended county level EAE-GDP results are closer to the EMDAT estimated AAL than the defended city level EAE-GDP results. However, EM-DAT data reflects per event losses whereas this study considers static losses for assuming floods for
different return periods occur across the entire area. This will ultimately lead to much higher expected annual losses for the exceedance probability curves derived from this hazard map comparison. This also highlights that the county level protection layer is not precise enough and leads to an underestimation of EAE and therefore is less suited for conclusions on the overall risk (i.e. the EAE) than more detailed (even if incomplete within the model domain) defense information.

### 4.4 Model Agreement

The model agreement maps shown in Figures 2a-b depict the model agreement at the grid cell level for undefended fluvial and combined hazard maps for RP100. For fluvial hazard maps, the areas with highest model agreement are mainly situated next to large rivers or deltas in eastern and northwestern China. A similar result is found for the combined flood hazard map of Figure 2b. Comparing the results of both flood type hazard maps, it appears that the combined flood hazard maps (Figure 2b) have higher model agreement for these flood hotspots. Furthermore, the combined hazard maps show an increased level of
detail due to higher native output resolutions. An overview of the model agreement index (MAI) for the whole of China is provided in Table 5.





The MAI scores for RP100 are 0.29 for the fluvial hazard maps and 0.38 for the 3 combined hazard maps. The change in MAI between RPs is the largest between RP20(-25) and RP50 for both undefended flood type hazard maps and becomes slightly smaller at higher RPs. Comparing the results of the undefended and county level defended hazard maps, the defended hazard

maps have lower MAI scores for both flood types below RP500, and there is no difference between MAI scores for the defended and undefended maps at RP500 and above as no flood defenses are in place. The city-scale defended hazard maps are not included in the MAI results section due to the small change in inundated area and therefore model agreement.

For both combined and fluvial hazard maps, MAI scores increase from low to high RPs. Model disagreement occurs mainly at the floodplain edges and on the modelling of smaller streams and rivers due to differences in considered catchment size of

the GFMs. This effect is more pronounced for smaller RPs.

The average MAI scores on a province level shown in Figures 4a-b show spatial differences of model agreement in China. MAI scores are higher (0.30-0.60) in the northwestern and eastern provinces for the fluvial hazard map in Figure 4a. The same map shows that model agreement is low in western China, the provinces in the south, and especially the Island of Hainan, with MAI scores between 0.10 and 0.30. The combined hazard map results in Figure 4b show a different spatial distribution of MAI

scores. The scores are highest in the northern provinces (0.50-0.65), some of the southern provinces (0.50-0.55), and the eastern provinces (0.55-0.60). The delta areas in the eastern and northeastern regions, and the provinces in western China, have lower MAI scores (0.35-0.50) than the previously mentioned regions.

These results indicate the importance of modelled catchment size and output resolution of the GFMs for the hazard maps. For example, the fluvial hazard maps of the JRC model only include catchments larger than 5000 km$^2$, while the Fathom model

includes catchment sizes of 50 km$^2$ and larger for their fluvial hazard maps. Therefore, model agreement will be low between these two, simply because JRC hazard maps do not model the inundation in headwater catchments or smaller catchments. This can be seen from the low MAI score for the relatively small island of Hainan in the south of China, which is not modelled by all GFMs. The higher MAI scores in the deltas of China for the fluvial hazard map are in line with expectations, because the inundated area is large and therefore a higher 'hit rate' between models is more likely. Not in line with expectation is that the

fluvial and combined flood type hazard maps show differences in the spatial distribution of model agreement. A likely reason for the combined hazard maps having higher model agreement in the mountainous parts of China is again the similarity in modelled catchment size and output resolution. For the end-user the higher MAI in these regions demonstrates more robustness in results and therefore shows that the selection of GFM should be considered based on the location of interest.

### 4.5 Limitations

The comparison of flood hazard maps is based on flood extent, where every grid cell is considered as fully inundated at more than 0 cm of flood depth. In this study we did not test the effect of this assumption on the results. A possible effect is the overestimation of flood extent by coarse resolution models as for example a grid cell with a small amount of inundation can be disaggregated to multiple inundated grid cells and therefore misrepresent the native flood hazard maps. A future study





would benefit from testing multiple inundation thresholds for converting flood depth to flood extent or by adding methods to
compare inundation depth

An additional limitation is the lack of RPs, especially the lower RPs, that shape the EAE-GDP results. Linear extrapolation of exposed GDP results to RP5 can misrepresent how GFMs simulate low RP floods. This affects the EAE-GDP, because the results of low RP floods have a larger weight on the results than high RP floods. Future studies should test multiple extrapolation and or interpolation methods.

Our study has focused solely on the inter-comparison of the outputs of the 8 GFMs and has not attempted a validation against past flood event footprints or results of regional flood maps. Therefore, results can currently only be interpreted relative to one another.

## 5 Conclusions and Outlook

In this study, we compared the flood hazard map outputs of GFMs for the country of China. The main aim was to carry out a
comprehensive comparison of flood hazard maps from 8 GFMs for the country of China, and assess how differences in the simulated flood extent between the models lead to differences in simulated exposed GDP and expected annual exposed GDP. We advance on past work by including pluvial flooding and the flood defenses in the comparison.

The main findings of this study are:

-    There are large variations between the hazard map outputs of GFMs in terms of inundated area and exposed GDP
490          percentages of China.

-    Similar results were found by Trigg et al. (2016) for the African continent. The difference in the CIMA-UNEP model results between these studies underlines the importance of the native output resolution of the flood hazard maps.

-    Higher model agreement is found for the combined hazard maps that include pluvial flooding than for fluvial hazard maps. This is due to greater similarity in the native output resolution and the considered catchment size of the three
495          models (Fathom, JBA, and KatRisk) that include pluvial flooding. The spatial distribution of model agreement differs between both types of flood hazard maps.

-    Pluvial flooding (both flooding of headwater catchments and off floodplain flooding) is a highly important type of flooding (for China). Depending on the minimum catchment size used for modelling fluvial floods, adding pluvial flooding can increase the expected annual exposed GDP by as much as 1.3% points.

-    Incorporation of external flood protection standards in the flood hazard maps reduces the variability of inundation and exposed GDP percentages between GFMs. Knowledge of structural defenses in high exposure areas is key in adequately assessing the overall risk of a country. County level (policy level) defense knowledge can help to further improve the results, but needs careful checking.

- The inclusion of industry models that currently model flooding at higher resolution both on the grid as well as on the catchment level and that additionally include pluvial flooding component strongly improved the inter-model comparison and provides important new benchmarks for the exposure at risk.

GFMs are complex modeling chains, with assumptions and uncertainties in the data used, the individual model components and their parameterization. In our study we can draw some inferences on the impact of certain modelling decisions on the hazard map output; however, we cannot conclude on GFM quality or the quality of an individual model component. In order to get a better understanding of which of these characteristics has the largest influence on overall model performance, a systematic comparison framework is required, in which each of these modeling components and parameters can be tested individually and in unison. The proposed model comparison framework of Hoch and Trigg (2019) could therefore greatly benefit our current understanding of global flood hazard.

In the future, multiple improvements are expected that can greatly benefit GFMs and their use for risk assessment. On the front of climate reanalysis datasets the successor of ERA-Interim, ERA5, has been released, increasing spatial and temporal resolution among other aspects. To improve accuracy of flood hazard maps there is a demand for next generation DEMs that will increase modelling resolution, result in better parameterization of hydrodynamic modelling and has the potential for capturing flood defenses. Improvements on current DEMs have been made by the creation of the Merit DEM (Yamazaki et al., 2019) that better captures among others the river networks.

For a complete risk assessment it is of importance to include coastal flooding to complete the flood mapping (Couasnon et al., 2019). Including climate change effects in the creation of flood hazard maps will further improve the representativeness of return period floods.

**Acknowledgement**

PJW received funding from the Dutch Research Council (NWO) in the form of a VIDI grant (grant no. 016.161.324).

**Author contribution**

JPMA, SU and PJW conceived the study. All co-authors contributed to the development and design of the methodology. JPMA analyzed and prepared the manuscript with contributions from all co-authors.

**Competing interest**

The authors declare no conflict of interest.



**Code/Data availability**

Code used for analyse is available at https://github.com/jeromaerts/flood_hazard_map_comparison_2019

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

**Table 1:** Technical specification of the flood hazard maps of the 8 GFMs.

*1 CIMA-UNEP: defenses are readily built-in, see section 3.5 for more information.
*2 Fathom: Defended hazard maps contained very limited flood defenses, not included in this study.
*3 JBA: Includes not readily built-in flood protection layer, see section 3.5 for more information.

| Global Flood Model | Flood Type | Return Periods | Output Resolution |
|---|---|---|---|
| CaMa-UT | Fluvial | 5, 10, 20, 25, 50, 100, 200, 250, 500, 1000, | 18 arc seconds |
| CIMA-UNEP[*1] | Fluvial | 25, 50, 100, 200, 500, 1000, | 32 arc seconds |
| ECMWF | Fluvial | 5, 20, 25, 50, 75, 100, 500, 1000, | 18 arc seconds |
| Fathom[*2] | Fluvial + Pluvial | 5, 10, 20, 25, 50, 75, 100, 200, 250, 500, 1000, | 3 arc seconds |
| JRC | Fluvial | 10, 20, 50, 100, 200, 500 | 30 arc seconds |
| GLOFRIS | Fluvial | 5, 10, 25, 50, 100, 250, 500, 1000, | 30 arc seconds |
| KatRisk | Fluvial + Pluvial | 10, 20, 50, 100, 200, 500, 1500 | 3 arc seconds |
| JBA[*3] | Fluvial + Pluvial | 20, 50, 100, 200, 500, 1500 | 1 arc second |


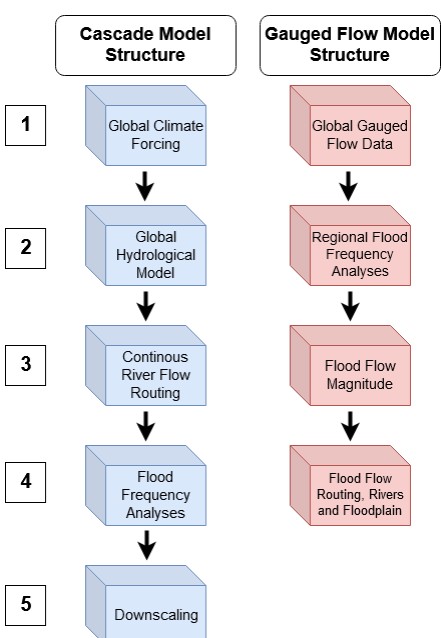

**Figure 1:** Two types of model structures as introduced by Trigg et al. (2016), with the cascade model structure in blue and the gauged flow model structure in red.





**Table 2:** Summary of the main model characteristics of the 8 GFMs.

| GFM/ Model Characteristics | CaMa-UT | CIMA-UNEP | ECMWF | Fathom |
|---|---|---|---|---|
| Model Type | Cascade Model | Gauged Flow Model | Cascade Model | Gauged Flow Model |
| Flood Type | Fluvial | Fluvial | Fluvial | Fluvial + Pluvial |
| Input Dataset | JRA-25 | EC-Earth /GRDC | ERA-Interim | GRDC + USGS |
| Global Hydrologial Model | MATSIRO-GW | Continuum Model | HTESSEL | N/A |
| River Routing Model | CaMa-Flood | Simplified Hydraulic Model | CaMa-Flood | LISFLOOD-FP |
| River Routing Type | Complex 2D Sub-Grid | Simple 1D | Complex 2D Sub-Grid | 2D Hydrodynamic |
| Digital Elevation Model | SRTM 3 | SRTM 3 | SRTM 3 | SRTM 3 |
| Considered Catchment Size | 500 km2 | 1000 km2 | 500 km2 | 50 km2 |
| Modeled Resolution | 3 arc seconds | 3 arc seconds | 3 arc seconds | 3/ 30 arc seconds |
| Output Resolution | 18 arc seconds | 30 arc seconds | 18 arc seconds | 3 arc seconds |
| **GFM/ Model Characteristics** | **JRC** | **GLOFRIS** | **KatRisk** | **JBA** |
| Model Type | Cascade Model | Cascade Model | Gauged Flow Model | Gauged Flow Model |
| Flood Type | Fluvial | Fluvial | Fluvial + Pluvial | Fluvial + Pluvial |
| Input Dataset | ERA-Interim | EU-WATCH | CPC + ERA-Interim | CRU TS3.2 + CFSRv2 + Local Data |
| Global Hydrologial Model | HTESSEL | PCR-GLOBWB | TOPMODEL modified | N/A |
| River Routing Model | LISFLOOD-Global | DynRout | Unit Hydrographs | RFlow/ JFlow |
| River Routing Type | 2D Hydrodynamic | 2D Volume | 2D Hydrodynamic | 1D + 2D Simple/ 2D Hydrodynamic |
| Digital Elevation Model | SRTM 3 | SRTM 3 | SRTM 3 | NEXTMAP World30 DSM/ Bare Earth DTM |
| Considered Catchment Size | 5000 km2 | Strahler order >= 6 | >4 cm Flood Depth | No Minimum |
| Modeled Resolution | 30 arc seconds | 30 arc seconds | 3 arc seconds | 1 arc second |
| Output Resolution | 30 arc seconds | 30 arc seconds | 3 arc seconds | 1 arc second |


**Table 3:** MAI calculation based on an example grid with a river indicated in bold with value 0.

| | | | | | | Example Calculation |
|---|---|---|---|---|---|---|
| 1 | 2 | 4 | **0** | 3 | 3 | N = 5 (Amount of Models) |
| 1 | 2 | 4 | **0** | **0** | **0** | A = 23 (Maximum Inundated Area) |
| 2 | 4 | **0** | 4 | 4 | 4 | a2 = 6 |
| 4 | **0** | 4 | 2 | 2 | 1 | a3 = 2 |
| **0** | 4 | 2 | 1 | 1 | 1 | a4 = 9 |
| | | | | | | MAI = (2/5*6)+(3/5*2)+(4/5*9)/23 |
| | | | | | | **MAI = 10.8/23 = 0.47** |


**Figure 2:** Aggregated flood hazard maps for both flood types, where the numbers and corresponding colors indicate the number of models
in agreement on the inundation of a grid cell. **2a**: Aggregated undefended fluvial flood hazard maps of 7 GFMs for RP100. **2b:** Aggregated
undefended combined (fluvial and pluvial) flood hazard maps of 3 GFMs for RP100.

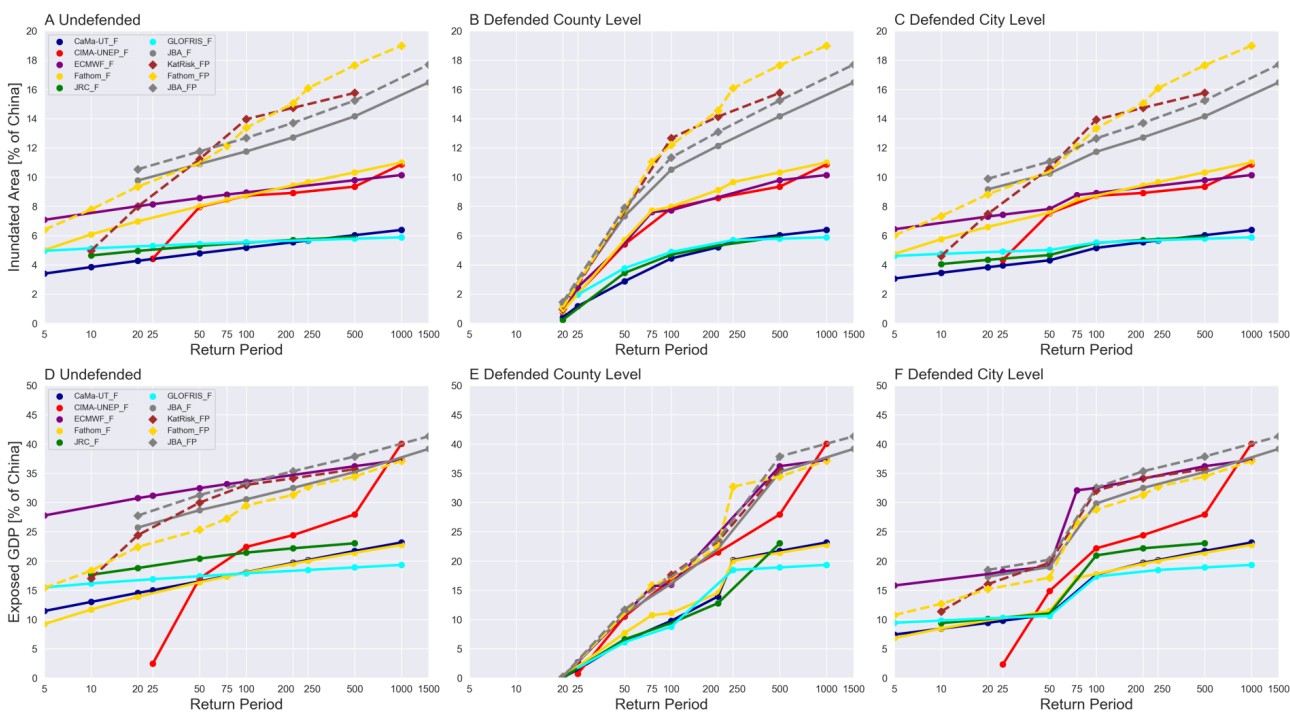

**Figure 3:** Results of multiple return period fluvial and combined hazard maps of 8 GFMs. The results of the fluvial hazard maps (_F) are represented by a continuous line and those of the combined hazard maps (_FP) by an interrupted line. The RPs range from 5 to 1500 and are displayed on a logarithmic horizontal axis. **3a:** Percentage of inundated area of China of undefended fluvial and combined hazard maps. **3b:** Percentage of inundated area of China of county level defended fluvial and combined hazard maps. **3c:** Percentage of inundated area of China of city level defended fluvial and combined hazard maps. **3d:** Exposed GDP percentage of China of undefended fluvial and combined hazard maps. **3e:** Exposed GDP percentage of China of county level defended fluvial and combined hazard maps. **3f:** Exposed GDP percentage of city level defended fluvial and combined hazard maps.




**Table 4:** EAE-GDP results of the 8 GFMs for the undefended, county level defended, and city level defended exposed GDP scenarios. The values are expressed as EAE-GDP percentages of China.

| Global Flood Model | Flood Type | Undefended EAE-GDP (% total GDP) | Defended County Level EAE-GDP (% Total GDP) | Defended City Level EAE-GDP (% Total GDP) |
|---|---|---|---|---|
| CaMa-UT | Fluvial | 2.34 | 0.07 | 1.56 |
| CIMA-UNEP | Fluvial | 0.53 | 0.19 | 0.50 |
| ECMWF | Fluvial | 5.59 | 0.10 | 3.26 |
| Fathom | Fluvial | 1.91 | 0.08 | 1.44 |
| JRC | Fluvial | 3.56 | 0.11 | 1.98 |
| GLOFRIS | Fluvial | 3.10 | 0.36 | 1.93 |
| JBA | Fluvial | 5.14 | 0.10 | 3.48 |
| KatRisk | Combined | 3.55 | 0.19 | 2.47 |
| Fathom | Combined | 3.18 | 0.12 | 2.28 |
| JBA | Combined | 5.55 | 0.10 | 3.71 |

5    **Table 5:** MAI results for the undefended and county level defended fluvial and combined hazard maps for multiple RPs.

| Number of GFMs | Flood Type | Return Period | Undefended MAI (-) | County Defended MAI (-) |
|---|---|---|---|---|
| 7 | Fluvial | 20-25 | 0.26 | - |
| 7 | Fluvial | 50 | 0.28 | 0.26 |
| 7 | Fluvial | 100 | 0.29 | 0.28 |
| 7 | Fluvial | 500 | 0.29 | 0.29 |
| 3 | Combined | 20 | 0.35 | 0.23 |
| 3 | Combined | 50 | 0.37 | 0.34 |
| 3 | Combined | 100 | 0.39 | 0.38 |
| 3 | Combined | 500 | 0.41 | 0.41 |
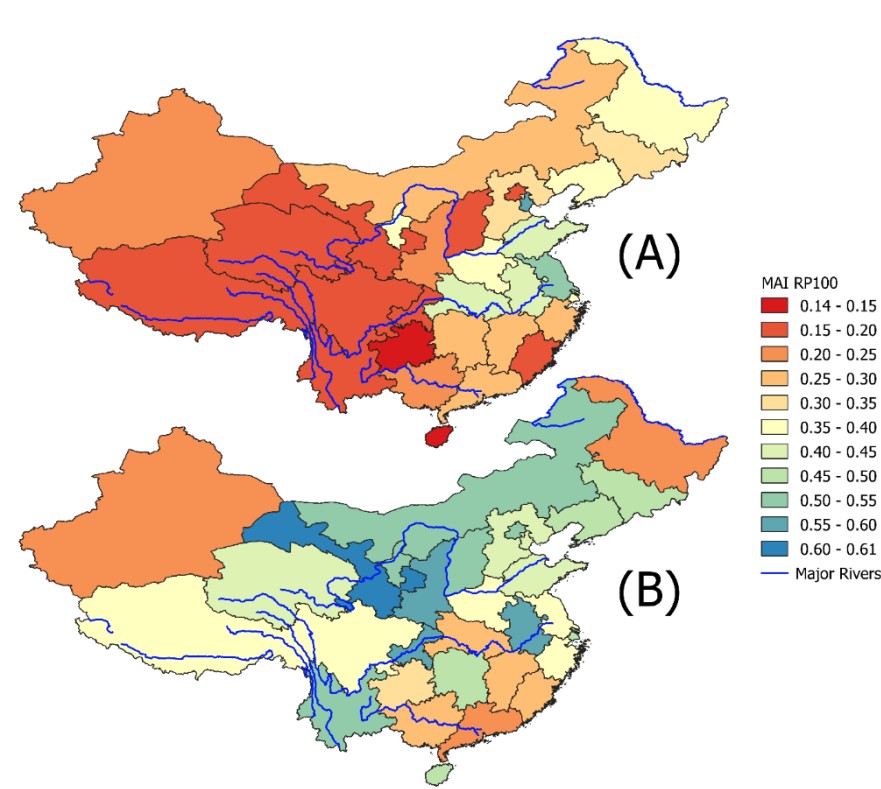

10  **Figure 4:** The spatial distribution of average MAI results on a province level for RP100 in China. **4a:** Undefended fluvial hazard map MAI scores (7 GFMs). **4b:** Undefended combined hazard map MAI scores (3 GFMs).