# Peer review of "Comparison of global flood model estimates of flood hazard and exposed GDP: a China case study"

_Natural Hazards and Earth System Sciences, 2020_

## Referee Comment (RC1) · Anonymous Referee #1 · 27 Mar 2020

The paper presents a very interesting comparison between the main global flood models used in recent studies to assess exposure to flood risk on a global scale. In my opinion, it makes a very interesting contribution towards the understanding of the impacts of various model assumptions on local and global risk descriptors.

The paper may represent a fundamental starting point for the future innovation of the modeling chains necessary for the assessment of hydraulic risk on a global scale. I'm therefore convinced that the paper should be published with a minor review, simply completing the conclusions.

The paper highlights (page 11) the importance of the impact of climate forcings on the results of the entire chain, a comment that I consider extremely interesting. It should, in my view, also be highlighted in the conclusions. Furthermore, the importance of the

size of the modeled basins, asin addition to the native resolution of the hazard map (already cited in the conclusions), could also be better commented in the conclusions.

Minor comments page 14, row 434 change numbering to 4.5 pag 15 row 469 change, accordingly, numbering to 4.6

Conclusions, page 16, row 491 it may be correct to quote again the work of Rudari et al. 2015, which comes to similar results
* * *

---

## Referee Comment (RC2) · Dominik Paprotny (Referee) · 15 May 2020

The manuscript "Global flood hazard map and exposed GDP comparison: a China case study" is a comparison of flood hazard maps computed from eight different global models. In general the paper is well written and covers an important and interesting topic. However, its main problem is that the paper advances our knowledge very little compared to previous studies in the field, including those cited by the authors. I think it is partially because the abstract, introduction and conclusions do not make the contribution convincing enough. More specifically:

1) The authors analyse differences between models purely descriptively. There are no real conclusions from these extensive descriptions. The models differ greatly even

at basic elements such as input and output resolution and minimum catchment size, which were not homogenized. For smaller return periods, the flood defence assumptions are much more relevant than any difference between models. And that is only the hazard component, without even considering uncertainty in exposure. At least some comparison between different model set-ups would be revealing on the sensitivity of the models. I understand that the authors only had access to output maps, however something could be done with GLOFRIS, as it is the authors' in-house model.

2) I recall from the recent EGU conference that the authors' argument for the study, when asked how does it differ from Trigg et al. (2016), was that it includes "industry" models. Their addition doesn't change much here, because despite a rather close agreement between them, there is no indication given whether they are any closer to actual flood hazard or risk. I think to should be better explained, because the argument in the paper is basically

3) Study area: the authors write "China is selected as our case study area because it poses many challenges to flood modelling: data scarcity; a variety of flood mechanisms spanning many climatic zones; complex topography; strong anthropogenic influence on the flood regimes, for example through river training; and a very high concentration of exposure." Firstly, why data scarcity would be the reason for choosing this case study? This only reduces the chances that a sensible comparison with some detailed local flood studies could be made. The study doesn't even try to establish whether any of the models is close to plausible hazard or risk levels (in contrast to Bernhofen et al. 2018). I would rather expect that a data-rich region (with rather good terrain, climate, flood protection, exposure, and historical floods data) would be better, by comparing how the global models deviate from dedicated, high-resolution studies for that area. On the data side, China seems only to offer a (rough) flood protection dataset. The remaining arguments of the authors are pretty much valid for any large country or subregion of any continent. A better explanation is definitely needed.

Overall, the lack of novel conclusions, the poor choice of the study area and a nonsystematic comparison of the models (more appropriate for a review article) are the weak points of the paper. I am sure that the authors will be able to successfully revise their paper with more convincing rationale for the study framework and by extracting stronger, practical conclusions from the analysis. I would be looking forward to read the revision.

Some smaller things I noted down while reading:

Title: I find the title rather confusing, because it is difficult to say what is compared with what. It definitely doesn't make it clear that the different global flood models are compared in terms of exposed GDP for China. It makes rather an impression that a single global map is analysed and rather multiple GDP datasets are used. I suggest to reformulate to clarify that many flood hazard models are compared with each other.

Abstract: The abstract is too long, and consequently lacks enough impact. The first two sentences should be removed (also because no citations should be made in the abstract unless strictly necessary, and simply repeat the introduction). The remaining parts on the introduction and methods are alright, while the results should be more compressed with more concise sentences.

L63: 'GHM' is not defined.

L104-L114: the paragraph lacks a reference to Table 1. The reference to section 2.5 – it should probably be 3.5. "The flood hazard maps are undefended" – please rewrite this in a way that sounds less... weird. Please define here what are the "undefended" and "defended" maps. The text & tables (Tab. 1 & 2) indicate different resolutions of CIMA model – which one is correct?

Eq. 1; what is the term i?

L291: should be $P = 1/R$ rather than $1/RP$. A single symbol should be used per variable (note that you use P to mean two different things); same for eq. 1 (EAE) and 2 (MAI).

Fig. 2: "Major rivers" should be removed, as they only obscure the flood zones. The

map lacks a grid and scale.

Fig. 4: provincial boundaries are not indicated in the legend. No scale.

Section 4.4-4.5: numbering is repeated.

The comparison between EAE and EM-DAT observed loss (4.4) is largely pointless as the authors (1) do not consider influence of the uncertainty in exposure estimates, which has pronounced effect on losses (2) consider only exposure of GDP, and not assets, which are worth many times more (4.5 times in China in 2015, according to Penn World Table) (3) do not use damage functions which transform exposure into losses. I think this whole subsection can be removed.

L484-5: The first part of the second sentence basically repeats the previous sentence.

L506: "exposure at risk" – exposure by definition involves something being "at risk". "flood exposure" would suffice in this sentence.

Additionally, I would suggest to check the conclusions in terms of writing, because while most of the paper is written well, it strangely drops in quality towards the end. E.g.: "…exposed GDP percentages of China", "County level (policy level) defense knowledge…", "For a complete risk assessment it is of importance to include coastal flooding to complete the flood mapping".

---

## Author Comment (AC1) · 1 Jul 2020

**Supplementary to Review #1**

Dear Reviewer,

Thank you for your constructive comments. Please find below our point by point reply to your comments and how we have adjusted the paper accordingly.

1) The paper highlights (page 11) the importance of the impact of climate forcings on the results of the entire chain, a comment that I consider extremely interesting. It should in my view, also be highlighted in the conclusions. Furthermore, the importance of the size of the modelled basins, as in addition to the native resolution of the hazard map (already cited in the conclusions), could also be better commented in the conclusions.

We agree that these model characteristic-based conclusions are one of the valuable insights that this study provides. We therefore reworked section 5 (Conclusions & Outlook) so that it better highlights these findings. The new text can be found on p. lines 496-498. In addition, we improved the text quality of this section.

2) Minor comments page 14, row 434 change numbering to 4.5 page 15 row 469 change, accordingly, numbering to 4.6.

We adjusted the numbering based on your suggestion.

3) Conclusions, page 16, row 491 it may be correct to quote again the work of Rudari et al. 2015, which comes to similar results.

Thank you - we have added the reference to Rudari et al. (2015) in section 5 (Conclusions & Outlook, p. 16, line 495).

---

## Author Comment (AC2) · 1 Jul 2020

**Supplementary to RC #2**

Dear Dominik Paprotny,

Thank you for your constructive comments. Please find below our point by point reply to your comments and how we have adjusted the paper accordingly.

1) In general the paper is well written and covers an important and interesting topic. However, its main problem is that the paper advances our knowledge very little compared to previous studies in the field, including those cited by the authors. I think it is partially because the abstract, introduction and conclusions do not make the contribution convincing enough.

We are glad that the reviewer finds the paper to be well written, and on an important and interesting topic. We would like to begin by stating that we agree with your suggestion to clarify the novel contribution in the abstract, introduction and conclusions, and have thoroughly revised these sections to do that.. Advancing the knowledge about global flood models and the use and credibility of their output (hazard maps) for flood risk assessment is an objective of our study, and indeed there is already an existing body of studies with a similar objective. Still, we believe that we advance beyond those studies in several ways, but these have not been articulated well enough. Please see our replies to your specific comments 1 and 2 on how we have taken care of this.

*1) "The authors analyse differences between models purely descriptively. There are no real conclusions from these extensive descriptions. The models differ greatly even at basic elements such as input and output resolution and minimum catchment size, which were not homogenized. For smaller return periods, the flood defence assumptions are much more relevant than any difference between models. And that is only the hazard component, without even considering uncertainty in exposure. At least some comparison between different model set-ups would be revealing on the sensitivity of the models. I understand that the authors only had access to output maps, however something could be done with GLOFRIS, as it is the authors' in-house model."*

We understand your argument but need to highlight that we did not to attempt a sensitivity analysis, which would systematically vary the inputs, parameter spaces, and potentially components of one model to understand the impact on a target output variable. Nor did we attempt a validation study like Bernhofen et al (2018) to understand the absolute differences between observed and modelled hazard. Rather, we are conducting this study from a pure practitioners perspective, e.g. the persona of a flood risk analyst who – in the absence of good validation points (like flood event footprints, authority hazard maps) and for a large geographical space – needs to understand the range at which the output of several models (here hazard maps) varies and what this implies for the choices in the next steps of developing a risk assessment (or full flood risk model) and its inherent uncertainties. The purpose of this study is therefore a) to assess the relative differences in the hazard output of a wide variety of global flood models, b) to understand and explain these differences from the differences in the models themselves (data, methods, modelling and output resolution), and c) to provide a simple analysis on the impact of these differences to flood risk. We are well aware of the added levels of uncertainty for a full flood risk analysis related to the modelling of vulnerability and exposure and discuss this in section 4.6 (Limitations) of the paper. The new text can be found on p. 16, lines 481-484. Based on your suggestion we have improved the abstract, introduction, and conclusions accordingly.

We do believe that our conclusions based on extensive model descriptions do not only provide new insights on the current state of GFMs (by including "industry" models) but also show aspects of current validation studies that are lacking. For example, the inclusion of pluvial flooding and the

presence of structural flood defences is not yet taken into account in most validation studies. By demonstrating the importance of these components we contribute to the quality of future validation studies.

*2) I recall from the recent EGU conference that the authors' argument for the study, when asked how does it differ from Trigg et al. (2016), was that it includes "industry" models. Their addition doesn't change much here, because despite a rather close agreement between them, there is no indication given whether they are any closer to actual flood hazard or risk. I think to should be better explained, because the argument in the paper is basically.*

The complete answer to how our study differs from Trigg et al. (2016) is that we include "industry" models, the pluvial flood component, and the presence of structural flood defences in our study area. Although "industry" models and "academic" models have been developed in parallel there is limited information available on how these "industry" models are setup, let alone compare to "academic" models. Being the first to include these models we believe that we lifted a curtain that allows the reader to get a better sense of how these "industry" models are set-up and how they compare to "academic" models. In particular, what we can demonstrate is that the commercially available models have generally advanced the global flood modelling by modelling much smaller catchments than all other global flood models and, in doing so, allow to account for pluvial flooding which is a major contributor to flood risk globally. Given the differences in methods used by the three commercial methods it is of high importance to note that they eventually "cluster" in their results (section 4.2, Figure 3, highlighting a) that there is some consistency in the results and b) the importance and relative contribution of pluvial flooding to flood risk (for China) and that excluding this type of flooding will likely lead to gross underestimation of the actual flood risk.

*3) Study area: the authors write "China is selected as our case study area because it poses many challenges to flood modelling: data scarcity; a variety of flood mechanisms spanning many climatic zones; complex topography; strong anthropogenic influence on the flood regimes, for example through river training; and a very high concentration of exposure." Firstly, why data scarcity would be the reason for choosing this case study? This only reduces the chances that a sensible comparison with some detailed local flood studies could be made. The study doesn't even try to establish whether any of the models is close to plausible hazard or risk levels (in contrast to Bernhofen et al.2018). I would rather expect that a data-rich region (with rather good terrain, climate, flood protection, exposure, and historical floods data) would be better, by comparing how the global models deviate from dedicated, high-resolution studies for that area. On the data side, China seems only to offer a (rough) flood protection dataset. The remaining arguments of the authors are pretty much valid for any large country or sub region of any continent. A better explanation is definitely needed.*

The key advantage of global flood models is their ability to create flood hazard maps that span data scarce regions. By choosing a region with the mentioned flood modelling challenges we provide an unbiased testbed for comparing these models which does not favour one specific model. Again, the case study was specifically chosen to reflect a real world problem that practitioners face – assessing flood risk in a region that is a) of high importance for flood risk assessment due to a combination of high flood potential and high concentrations of exposure which is highlighted in a long history of both severe and frequent flood losses, b) of large geographical extent that makes it hard to work with just a few validation points, and c) for which it is hard to access data. Indeed, an important next

step would be to follow up this kind of research with in-depth validation studies, similar to how Bernhofen et al. built on the initial comparison study by Trigg for the continent of Africa.

We have reworked section 1 (Introduction) accordingly to make our choice for China as a case study clearer.

Our response to the more minor comments.

- Title: We changed the title to "Comparison of global flood model estimates of flood hazard and exposed GDP: a China case study", such that it better describes what we compare.
- Abstract: We shortened the abstract as per your suggestion.
- Results: We compressed the results by removing repetitive explanations and made the sentences more concise.
- EAE and EM-DAT comparison: We dropped this part of the results based on your suggestion. The reason for providing this comparison in the first place was to add context to the results. We were aware that a one on one comparison would be based heavily on assumptions and have therefore removed this comparison.
- Conclusions: We agree with your findings and improved the quality of the text.
- Figures: We added a scale to the figures, updated the legend, and increased the transparency of the large rivers.